# A PLLA Coating Does Not Affect the Insertion Pressure or Frictional Behavior of a CI Electrode Array at Higher Insertion Speeds

**DOI:** 10.3390/ma15093049

**Published:** 2022-04-22

**Authors:** Dana Dohr, Katharina Wulf, Niels Grabow, Robert Mlynski, Sebastian P. Schraven

**Affiliations:** 1Department of Otorhinolaryngology, Head and Neck Surgery “Otto Körner”, Rostock University Medical Center, 18057 Rostock, Germany; robert.mlynski@med.uni-rostock.de (R.M.); sebastian.schraven@med.uni-rostock.de (S.P.S.); 2Institute for Biomedical Engineering, Rostock University Medical Center, 18119 Rostock, Germany; katharina.wulf@uni-rostock.de (K.W.); niels.grabow@uni-rostock.de (N.G.)

**Keywords:** PLLA coating, coated cochlear electrode array, insertion pressure, linear cochlear model, friction coefficient, insertion speed, insertion trauma

## Abstract

To prevent endocochlear insertion trauma, the development of drug delivery coatings in the field of CI electrodes has become an increasing focus of research. However, so far, the effect of a polymer coating of PLLA on the mechanical properties, such as the insertion pressure and friction of an electrode array, has not been investigated. In this study, the insertion pressure of a PLLA-coated, 31.5-mm long standard electrode array was examined during placement in a linear cochlear model. Additionally, the friction coefficients between a PLLA-coated electrode array and a tissue simulating the endocochlear lining were acquired. All data were obtained at different insertion speeds (0.1, 0.5, 1.0, 1.5, and 2.0 mm/s) and compared with those of an uncoated electrode array. It was shown that both the maximum insertion pressure generated in the linear model and the friction coefficient of the PLLA-coated electrode did not depend on the insertion speed. At higher insertion speeds above 1.0 mm/s, the insertion pressure (1.268 ± 0.032 mmHg) and the friction coefficient (0.40 ± 0.15) of the coated electrode array were similar to those of an uncoated array (1.252 ± 0.034 mmHg and 0.36 ± 0.15). The present study reveals that a PLLA coating on cochlear electrode arrays has a negligible effect on the electrode array insertion pressure and the friction when higher insertion speeds are used compared with an uncoated electrode array. Therefore, PLLA is a suitable material to be used as a coating for CI electrode arrays and can be considered for a potential drug delivery system.

## 1. Introduction

Cochlear implants (CIs) are used as the treatment of choice for severe to profound sensorineural hearing loss. With an increasing trend, more than 324,000 patients have received a cochlear implant worldwide [1]. During CI implantation, an electrode array is inserted into the scala tympani and restores the hearing of deaf patients through electrical stimulation of the spiral ganglion neurons (SGNs). However, aside from this advanced contribution, insertion of the electrode array causes trauma to the fragile endocochlear structures through mechanical irritation [2,3,4,5,6].

The trauma is characterized by the degeneration of the SGNs [7] and promotes the formation of fibrotic tissue or even leads to abnormal bone proliferations [8,9]. Additionally, the formation of fibrotic tissue is further supported by the electrode array, remaining as a foreign body in the cochlear duct [9].

The issue of how trauma is triggered has been the subject of many studies [10,11,12,13,14,15,16]. It was shown that insertion trauma increased with deep insertions [13] combined with electrode kinking [14] and also that the insertion force and insertion pressure play an important role in cochlear trauma development [15,16,17]. Furthermore, it was reported that the insertion force correlates with the friction between the electrode array and the endocochlear lining [18]. Moreover, the results of multiple studies indicate that the insertion force, pressure, and friction depend on the speed at which the electrode arrays are inserted [18,19,20,21].

Several methods have been developed to minimize the trauma from electrode insertion, including improvements in surgical techniques, electrode modifications, or the combination of both, such as partial electrode array insertion [11,22,23]. Despite further developments, endocochlear damage still cannot be completely prevented [11].

Therefore, in order to inhibit insertion trauma, the intra- and postoperative use of pharmaceutical agents is increasingly becoming a focus of research [24,25,26,27,28]. These agents are mainly glycocorticoids, which are administered locally or systemically [26,27,28]. A wide variety of systems are used to inhibit insertion trauma or to deliver drugs directly into the cochlea, such as intraoperative administration using catheters, longer-acting pump systems, or drug depots either in the silicone of the electrode array or in separate coatings [24,25,26,27,28,29,30,31,32,33,34]. The effect of polymer coatings without drug loading was tested by Hadler et al. [29]. In this study, it was shown that a polymer coating itself is a suitable option to block fibroblast overgrowth. However, few drug depots have been used clinically so far [24].

While drugs via catheters are administered as a bolus, pump systems steadily supply the cochlea over a longer period of time. However, pump cannula obstruction at low flow rates may occur, and microbiological control is required due to repeated transdermal refills [31]. In contrast, the drug depot is believed to be a safe method for cochlear application of corticosteroids [35,36] and reduces fibrosis in vivo [33]. As passive transport depends on the concentration gradient across the membrane, the kinetics follows Fick’s first law of diffusion [37]. 

A dual-drug delivery system which combined initial and long-term drug release with different kinetics was developed by Wulf et al. [38]. Here, the electrode array is covered by a drug-loaded poly-*L*-lactid *L*210 (PLLA) coating in addition to the drug-loaded silicone. Furthermore, it was demonstrated that electrical stimulation with active implants is not affected by the coating [38].

Nonetheless, how the coating impacts the mechanical properties of an electrode array has yet to be determined. Neither the insertion pressure of the coated electrode array nor the friction behavior of a PLLA-coated electrode array on the endosteum lining has been analyzed so far. The aim of this study was to investigate the insertion pressure of a PLLA-coated electrode array during placement into a linear cochlear model. Additionally, the friction coefficients at different movement speeds were evaluated between a PLLA-coated electrode array and a tissue that simulated the endosteum lining. All data were measured depending on different insertion speeds and were subsequently compared with uncoated electrodes.

## 2. Materials and Methods

### 2.1. Electrode Array

All experiments were performed with a straight standard insertion electrode array (MED-EL, Innsbruck, Austria) (Figure 1). The insertion part was 31.5 mm long and distinguished by proximal and distal ends with diameters between 0.5 and 1.3 mm, respectively. It contained 24 platinum electrode contacts separated by 2.4 mm of distance.

### 2.2. Electrode Array Coating

The general coating process was carried out according to Wulf et al. [38]. In brief, the cleaned silicone surfaces of the electrode arrays were first activated via O_2_ plasma using 100 W of power at 0.3 mbar for 1 min in a 2-L plasma chamber (Diener, Ebhausen, Germany). Then, the tips of the electrode arrays were incubated in pure GOPS for 4 h at 90 °C. The activated samples were rinsed 3 times with ethanol and dried at 80 °C overnight under a vacuum at 40 mbar. The coating of the activated electrode arrays was prepared via an established and characterized in-house manufactured spray-coating process. First, the tips of the electrode arrays were spray-coated with a thin polymer layer of PLLA-NH2 using a chloroform PLLA-NH2 (0.5 wt%) spray solution for 15 s with a rotation speed of 10 rpm and a flow of 0.2 bar. Afterward, the samples were dried at 80 °C overnight. Dried electrode arrays were coated with pure PLLA (chloroform PLLA solution (0.2 wt%)) in 10-µm thicknesses (measured via microscopy (Olympus SZX16, Olympus, Hamburg, Germany)). Regarding Wulf et al. [38], that lower layer thickness did not exhibit better impedance. The coating process ran for 300 s with a rotation speed of 10 rpm and a flow of 0.2 bar. Afterward, the samples were dried at 80 °C overnight under a vacuum. For scanning electron microscope (SEM) imaging, the surface of the electrode array was first sputtered with gold for better visualization and subsequently examined in a QUANTA FEG 250 (FEI Company, Dreieich, Germany) scanning electron microscope (SEM) (Figure 1c).

### 2.3. Insertion Pressure Measurements and Slope Calculation

#### 2.3.1. Artificial Linear Scala Tympani Model

The electrode arrays were inserted into a linear, artificial scala tympani model [39] (Figure 2a). This model was made of polyester and represented the average geometrical shape of an adult human scala tympani model, noting that the volume of the scala tympani was easily accessible along its entire profile. The model was separated into two parts: the uncoiled scala tympani (associated with the basal opening) and the extension intended for the pressure sensor (associated with the apical opening) (Figure 2a). The uncoiled scala tympani has the average length of an adult human scala tympani model duct (30 mm), and the basal opening has the average size of the round window of a cochlea [39]. The extension was 10 mm long, and the size of the apical opening was 0.5 mm. The model’s canal was lubricated with artificial perilymph. Perilymph was prepared by a mix of 145 mM NaCl (sodium chloride, Merck KGAA, Darmstadt, Germany), 2.7 mM KCl (potassium chloride, Merck KGAA, Darmstadt, Germany), 2 mM MgSO_4_ (magnesium sulfate, Merck KGAA, Darmstadt, Germany), 1.2 mM CaCl_2_ (calcium chloride, Merck KGAA, Darmstadt, Germany), and 5 mM HEPES (Merck KGAA, Darmstadt, Germany).

#### 2.3.2. Insertion Pressure Test Bench

The insertion pressure was measured with a micro-optical pressure sensor made by FISO (Quebec, Canada) using an FOP-NS-1048 sensor with fiber-type 62.5/125 gradiant glass. The sensor was mounted with alginate dental impression powder (Henry Schein Services GmbH, Gallin, Germany) and reached about 5 mm into the lumen of the linear cochlear model on the apical pole (Figure 2a). The sensor was run by Evolution software. The pressure was measured in mmHg, and the low pass filter frequency was set to 0.5 Hz. The test bench (Figure 2b) comprised two opposite holders. The upper holder was used to fix the electrode array and was driven by a linear feed. The lower holder was used to fix the linear scala tympani model adapted to the pressure sensor.

#### 2.3.3. Experimental Procedure

The electrode array was fixed as described above and inserted into the lubricated linear scala tympani model. The arrays were inserted into the entire model lumen over a distance of 30 mm under 5 different insertion speeds (0.1, 0.5, 1.0, 1.5, and 2.0 mm/s). The experiments were repeated 10 times using the same electrode array and insertion speed.

#### 2.3.4. Calculation of the Slope for the Insertion Pressure Curve

Since the pressure curve at the beginning of the CI insertion had great relevance for the chirurgic handling of the electrode, the increase in the insertion pressure was quantified. The increase was determined with the calculation of the slope. For this calculation, a start point (*X*_1_/*Y*_1_) and an end point (*X*_2_/*Y*_2_) value were defined for each pressure curve. The start point was set to the first value of the pressure measurement (at 0 mm) for all curves. The end point was defined where the difference between two y-values remained constantly minimal (≤0.1) over a distance of 5 mm. This represented the beginning of each individual start of the curve plateau. By defining the points, a fictitious straight line was generated from the start of the curve to its plateau, from which the slope could be calculated as follows:(1)$$m=\frac{y_{2}-y_{1}}{x_{2}-x_{1}}\;$$
where m is the slope of pressure increase, $x_{1}$ is the X-value of the start point, $y_{1}$ is the Y-value of the start point, $x_{2}$ is the X-value of the end point, and $y_{2}$ is the Y-value of the end point.

#### 2.3.5. Calculation of the Area under the Curve for the Insertion Pressure

The area under the curve indicates the pressure the cochlea was exposed to during the entire insertion. Hence, the AUC could be considered equivalent to continuous sonication of the inner ear and represents a meaningful parameter for evaluation. For each measured value, the area under the curve (AUC) was calculated with following formula:(2)$$AUC=\frac{(y_{1}+y_{2})\times \left(x_{2}-x_{1}\right)}{2}$$
where AUC is the area under the curve, $x_{1}$ is the X-value of the start point, $y_{1}$ is the Y-value of the start point, $x_{2}$ is the X-value of the end point, and $y_{2}$ is the Y-value of the end point.

From this, the sum of all calculated AUCs was determined and resulted in the pressure that arose over the entire insertion (30 mm).

### 2.4. Friction Measurements and Friction Coefficient Calculation

#### 2.4.1. Friction Partner and Lubricant

During CI implantation, the perilymph usually acts as a lubricant for the electrode array, and the endosteum lining of the scala tympani acts as a friction partner. Here, a porcine endothoracic fascia was used as a friction partner for mimicking the inner cochlear slippery conditions. The tissue was offal and kindly provided by a local farmer (Landfleischerei Wiechmann, Pankelow, Germany). It was taken from pigs up to 6 months old. Following slaughter, the fascia was carefully removed from the pigs’ ribs and stored in phosphate-buffered saline at 4 °C. Artificial perilymph was used as a lubricant (Section 2.3).

#### 2.4.2. Friction Test Bench

The friction forces were recorded by a modified pulling device [40], which has been described in detail previously by Dohr et al. [18] (Figure 2c). In brief, the friction test bench comprised two main parts: the fixed part and the mobile part. The fixed part consisted of a mount containing a deflection cylinder and a deflection pulley. The deflection cylinder was covered with the friction partner (endothoracic fascia) fixed by cable ties after cutting it into a 4 × 4 cm piece and being washed 3 times with artificial perilymph. The mobile part consisted of the electrode array, weight, a 5-N load cell, and the pulling device with a linear feed. The electrode array (Figure 1a) was proximally clamped to the weight and the distal to a wire rope with a diameter of 0.5 cm and length of 25 cm. Afterward, the array was wound over the deflection cylinder covered with the friction partner. The wire rope was deflected by the deflection pulley and connected to the load cell and pulling device with a linear feed. The weight attached on the proximal end of the array was chosen so that a weight force of 0.1 N acted on the array.

#### 2.4.3. Experimental Procedure

The electrode array and friction partner were fixed as described above. Subsequently, the electrode array was pulled over the friction partner using five different pulling speeds (0.1, 0.5, 1.0, 1.5, and 2.0 mm/s) representing the cochlear electrode insertion speed. The pulling distance was 30 mm along the electrode contacts. The fiction partner was constantly wetted with artificial perilymph, and the friction forces were recorded every 0.5 mm. For each speed, the friction partner was renewed, and three electrode arrays were used. Since the electrode array was under increased mechanical stress during measurement, the experiments were repeated only five times using the same electrode array.

#### 2.4.4. Friction Coefficient Calculation

Based on the impending slippage model [41] and the calculation performed by Kha et al. [42], the friction coefficient *µ* was determined using the following formula:(3)$$\mu =\frac{1}{\beta }\ln\frac{F_{friction}}{F_{G}}\;.$$
where β is the radiant of the friction partner covering the cylinder, *F_friction_* is the friction force recorded over a distance of 30 mm, and *F_G_* is the weight force (0.1 N) that acts on the electrode array.

### 2.5. Statistical Analysis

Data visualization and statistical analyses were performed with GraphPad Prism (version 8.02, La Jolla, CA, USA). Comparisons of multiple groups were performed by analysis of variance (ANOVA) followed by Bonferroni’s or Sidak’s multiple comparison post hoc test. The significance levels are indicated by “*” for *p* < 0.05, “**” for *p* < 0.01, “***” for *p* < 0.001, and “****” for *p* < 0.0001.

## 3. Results

### 3.1. Surface Morphology of Uncoated and PLLA-Coated Electrode Arrays

The uncoated electrode array and the results of the coating process reflected in the integrity of the PLLA coating were monitored by SEM images (Figure 1b,c).

The electrode array insertion part, which was distinguished into a proximal and distal end, is schematically illustrated in Figure 1a. It contained 24 platinum electrode contacts spaced by 2.4 mm of distance. Counted from the distal to proximal end, Figure 1b showed the fifth platinum electrode contact (hole) and the surrounding area of an electrode array before the coating process. Figure 1c represents the fifth electrode contact (hole) and the surrounding area after the coating process. The coating formed a thin film around the electrode body. The original shape was retained. The applied polymer provided a slightly structured surface, and fine cracks could be identified at the level of the electrode contacts (Figure 1c).

### 3.2. Impact of the Insertion Speed on the Electrode Insertion Pressure for Uncoated and Coated Electrode Arrays in the Linear Cochlear Model

To analyze the influence of the insertion speed on the insertion pressure of the uncoated and coated electrode arrays, the insertion pressure was measured in a linear cochlea model at different insertion speeds (0.1, 0.5, 1.0, 1.5, and 2.0 mm/s) (Figure 2a,b).

The insertion pressures of the uncoated electrode arrays were between −0.0036 and 1.2952 mmHg (Figure 3a). As shown in Figure 3a, the insertion pressure increased steeply at all insertion speeds until a plateau was reached. While the pressure plateau was reached after an insertion depth of 10 mm at insertion speeds of 0.5 and 1.5 mm/s, the insertion pressure at 1.0 and 2 mm/s rose faster, and the plateau was already reached at 6 mm and 8 mm. In contrast, the insertion pressure rose the slowest out of all speeds at an insertion speed of 0.1 mm/s. Here, the plateau was reached at 16 mm. The calculated slope of the pressure showed a significantly steeper increase in pressure at insertion speeds of 1.0 and 2.0 mm/s if compared with 0.1 mm/s (Table 1).

Considering the maximum insertion pressure of the uncoated electrode arrays (Figure 3b), it is shown that the lowest pressure was generated at an insertion speed of 0.1 mm/s (1.108 ± 0.096 mmHg). The pressure increased significantly as the speed increased if the data were compared to 0.1 mm/s. The highest maximum pressure occurred at a speed of 2.0 mm/s (1.363 ± 0.039 mmHg), followed by 1.5 mm/s (1.252 ± 0.034 mmHg) and 1.0 mm/s (1.195 ± 0.031 mmHg). Complementary to this, no significant difference was found between 0.1 and 0.5 mm/s (1.128 ± 0.056 mmHg).

For the analysis of how much pressure was generated during the distance while the uncoated electrode array was inserted into the cochlear model, the area under the curve (AUC) was calculated (Figure 3c). Here, the lowest pressure was generated at a speed of 0.1 mm/s (23.02 ± 3.87 mmHg × mm) and increased significantly at higher speeds. The highest pressure occurred at a speed of 2.0 mm/s (30.5 ± 1.76 mmHg × mm), followed by 1.0 mm/s (29.25 ± 1.63 mmHg × mm), 1.5 mm/s (28.12 ± 1.93 mmHg × mm), and 0.5 mm/s (26.83 ± 2.42 mmHg × mm).

Similar results were also obtained when analyzing the PLLA-coated electrode array (Figure 3d–f). The insertion pressures of the coated electrode arrays were between −0.0026 and 1.3278 mmHg (Figure 3d). As shown in Figure 3b, the insertion pressure increased steeply at all insertion speeds until they reached a plateau. While the plateau was reached at insertion speeds of 0.5 and 1.0 mm/s after an insertion depth of 6 mm and at 1.5 and 2.0 mm/s after 9 mm, the insertion pressure at 0.1 mm/s rose slower, and the plateau was reached at 14 mm. The calculated slope of the pressure showed a significantly steeper increase in pressure at insertion speeds of 0.5 and 1.0 mm/s if compared to 0.1 mm/s (Table 1).

The analysis of the maximum insertion pressure of the coated electrode arrays (Figure 3e) showed that there were no significant differences at the driven insertion speeds when the data were compared with 0.1 mm/s (1.266 ± 0.141 mmHg). The maximum insertion pressure was 1.241 ± 0.041 mmHg at an insertion speed of 0.5 mm/s, 1.283 ± 0.0632 mmHg at 1.0 mm/s, 1.268 ± 0.032 mmHg at 1.5 mm/s, and 1.353 ± 0.044 mmHg at 2.0 mm/s.

For the analysis of how much pressure was generated by a coated electrode array during the distance while the array was placed into the cochlear model, the area under the curve was calculated (Figure 3f). Here, the lowest pressure was generated at a speed of 0.1 mm/s (26.23 ± 4.41 mmHg × mm), increasing significantly at higher speeds. The highest pressure occurred at a speed of 0.5 mm/s (30.93 ± 1.59 mmHg × mm), followed by 1.0 (30.71 ± 1.93 mmHg × mm) and 2.0 mm/s (30.44 ± 2.698 mmHg × mm). Complementary to this, no significant difference was found between 0.1 and 1.5 mm/s (28.62 ± 2.09 mmHg × mm).

In conclusion, both the maximum pressure and the pressure during the insertion caused by an uncoated electrode array were significantly dependent on the insertion speed. On the other hand, for the PLLA-coated electrode array, the maximum pressure was not influenced by the insertion speed, but the pressure during insertion was.

### 3.3. Impact of PLLA Coating on Electrode Insertion Pressure

To determine the influence of the PLLA coating on the electrode array compared to an uncoated array, the maximum insertion pressure and pressure during insertion were compared.

Figure 4a shows the maximum insertion pressure of an uncoated versus a coated electrode array at different insertion speeds. It is demonstrated that the maximum pressures generated by the insertion of a coated electrode array at 0.1 (1.266 ± 0.141 mmHg), 0.5 (1.241 ± 0.041 mmHg), and 1.0 mm/s (1.283 ± 0.0632 mmHg) were significantly higher than the pressures generated by an uncoated electrode array (0.1 mm/s: 1.108 ± 0,096 mmHg, 0.5 mm/s: 1.128 ± 0.056 mmHg, 1.0 mm/s: 1.195 ± 0.031 mmHg). Interestingly, the insertion pressure converged at insertion speeds of 1.5 (coated vs. uncoated: 1.252 ± 0.034 mmHg vs. 1.268 ± 0.032 mmHg) and 2.0 mm/s (coated vs. uncoated: 1.363 ± 0.039 mmHg vs. 1.353 ± 0.044 mmHg). Here, the pressure did not differ significantly.

A comparison of the AUCs of the two types of electrode arrays (Figure 4b) showed that a significant difference only occurred at an insertion speed of 0.5 mm/s (uncoated vs. coated: 26.83 ± 2.41 mmHg × mm vs. 30.93 ± 1.59 mmHg × mm). In this case, the pressure during the insertion created by a coated electrode array was increased compared with an uncoated one at the same speed.

Taken together, the PLLA coating influenced the maximum insertion pressure at low speeds (0.1, 0.5, and 1.0 mm/s) and only influenced the pressure during insertion at an insertion speed of 0.5 mm/s.

### 3.4. Impact of the Moving Speed on the Friction Coefficient of Coated Electrode Arrays

To investigate the impact of the insertion speed on the coated electrode array friction, the friction forces between a coated electrode array and a tissue simulating the endosteum lining were recorded at different moving speeds (0.1, 0.5, 1.0, 1.5, and 2.0 mm/s). For better comparison of the recorded friction forces, the friction coefficients were calculated. Finally, the data were then compared with the friction coefficients of the uncoated electrode arrays to determine the effect of the PLLA coating on the friction properties of the electrode array.

Figure 5 underlines that the moving speed had no significant influence on the friction coefficient between the coated electrode array and the endosteum-simulating lining if compared with the friction coefficient that arose at 0.1 mm/s. The friction coefficients calculated on the basis of the frictional forces were 0.44 ± 0.12 at 0.1 mm/s, 0.43 ± 0.07 at 0.5 mm/s, 0.40 ± 0.15 at 1.0 mm/s, 0.51 ± 0.05 at 1.5 mm/s, and 0.46 ± 0.03 at 2.0 mm/s. Through comparison of the uncoated and coated electrode arrays (Figure 5b), the friction coefficient of the coated array significantly increased at speeds of 0.1 and 0.5 mm/s (uncoated vs. coated: 0.29 ± 0.12 and 0.30 ± 0.14 vs. 0.44 ± 0.12 and 0.43 ± 0.07), while negligible differences were found at 1.0, 1.5, and 2.0 mm/s (uncoated vs. coated: 0.36 ± 0.15, 0.54 ± 0.09, and 0.48 ± 0.13 vs. 0.40 ± 0.15, 0.51 ± 0.05, and 0.46 ± 0.03).

In summary, the obtained data indicate that the insertion speed had no effect on the friction coefficient of the PLLA-coated electrode array, and the friction data of the coated electrode array approached that of the uncoated array at higher insertion speeds (1.0, 1.5, and 2.0 mm/s).

### 3.5. PLLA Coating Condition after Pressure and Friction Measurements

To evaluate the PLLA coating condition after the pressure and friction measurements, SEM images were acquired from the selected coated electrode arrays.

Figure 6a–c shows a PLLA-coated electrode array before the ordered testing (control), and a PLLA-coated electrode array after the pressure (Figure 6d–f) and friction measurements (Figure 6g–i) is also shown. Furthermore, Figure 6a,b shows that the applied polymer provided a slightly structured surface and fine cracks that could be identified at the level of the electrode contacts. The tip of the electrode array was lightly covered with PLLA, and apart from minor irregularities, the PLLA coating did not peel off noticeably.

The PLLA coating of an electrode array that was inserted 10 times into a linear cochlear model at an insertion speed of 2.0 mm/s is shown in Figure 6d–f. Here, the coating at the level of the electrode contacts (Figure 6e) and between the contacts (Figure 6d) exhibited slight discontinuities, overlaps, and wrinkles. However, the coating at the tip of the electrode array (Figure 6f) was neither wrinkled nor peeled. Here, no obvious changes could be observed compared with the tip of the PLLA coated electrode array before experimental insertion (Figure 6c).

The PLLA coating of an electrode array that was used in the friction measurements is shown in Figure 6g–i. The electrode array was moved 5 times over a tissue, simulating the endosteum lining of the cochlea at moving speed of 1.5 mm/s. Similar to the control electrode array (Figure 6a), the coating between the electrode contacts did not show any noteworthy irregularities after the friction experiments (Figure 6g). In contrast, Figure 6h shows the deep cracks in the PLLA coating at the level of the electrode contacts. The cracks were deeper than those detected in the coating of the control electrode array (Figure 6b). Moreover, the coating at the tip of the electrode array was torn open and peeled off (Figure 6i).

In summary, after the pressure and friction experiments, the coating showed notable additional cracks and wrinkles compared with the coating of the control electrode array, especially at the level of the electrode contacts.

## 4. Discussion

This study reports on the mechanical properties of a PLLA-coated electrode array. Previously, no data on the biomechanical interaction of PLLA-coated electrode arrays for friction and insertion pressure were reported. Therefore, the insertion pressure of coated electrode arrays arising in a linear cochlear model and the friction data between those arrays and a tissue simulating the endosteum lining were measured and analyzed. All data were obtained at different insertion speeds and compared with the data from uncoated electrode arrays.

The current study shows that (1) the applied speeds (0.1–2.0 mm/s) had no significant impact on the majority of the investigated biomechanical properties (maximum insertion pressure and friction coefficient) of the PLLA-coated electrode arrays, and (2) there were no significant differences between the coated and uncoated electrode arrays in all investigated parameters at higher insertion speeds (from 1.0 mm/s onward).

While most of the known coatings for the CI electrode array tend to be made of hydrogel composites [33,34] or biocompatible 2-methacryloyloxyethyl phosphorylcholine (MPC) polymer [32], this is the first time that a PLLA coating has been used, which is well established in biomedical applications, such as in cardiovascular systems [43]. Here, a layer thickness of 10 µm was selected to ensure a sufficient surface area for future drug delivery while taking into account the small diameter of the vulnerable scala tympani. Impedance measurements of varied layer thicknesses of 2.5, 5, and 10 µm revealed that the lower layer thicknesses did not display superior impedances [38].

The shown SEM images (Figure 1b,c) revealed that the shape of the electrode array was not affected by the PLLA coating. It formed a slightly structured surface, which was the result of the hydrophobic properties of the PLLA and the coating process [38] (Figure 1c). In contrast, apart from the minor visible changes, the PLLA coating increased the diameter of the electrode array by 10 µm and caused the electrode array to lose flexibility. This inflexibility can be considered the reason for the indentations observed at the level of the electrode contacts (Figure 1c). Consequently, immobility was transferred to the coating and led to the formation of these cracks.

It is common practice to use a scale model of the cochlea to obtain mechanical data, and mostly soap is used as a lubricant [18,19,21,44,45]. In this study, however, a linear model was used, since the coated electrode could not be inserted into a scale model, and due to the inflexibility of the PLLA coating, even the model was lubricated with soap. Furthermore, artificial perilymph was used as a lubricant modeling more natural conditions. Since most insertion speed-related reports for cochlear implant studies have applied constant insertion speeds of 0.16–3.3 mm/s [17,19,46,47], a similar speed ranging from 0.1 to 2.0 mm/s was used in this study. Manually, the insertion technique and therefore the insertion speeds of the electrode arrays vary, depending on the surgeon. However, the goal should be to maintain a steady and notably slow insertion speed, as numerous studies have shown that the insertion speed affects the mechanical conditions and hence the CI insertion trauma [16,18,19].

Two parameters were used to evaluate the insertion pressure when an electrode array was inserted into the model: the maximum insertion pressure and the pressure during the insertion process. The maximum pressure gives information about what is the highest pressure the cochlea is exposed to during the insertion. Here, one value was considered at a certain time after a certain insertion distance. In contrast, the pressure during insertion is represented by the area under the curve (AUC). It indicates the pressure the cochlea is exposed to during the entire insertion. Hence, the AUC can be considered equivalent to continuous sonication of the inner ear and represents a meaningful parameter for evaluation.

When considering the progressions of the pressure curves (Figure 3a,d), the slowest insertion speed of 0.1 mm/s affected the insertion pressure of both the coated and uncoated electrodes. In these cases, the pressure increased the slowest compared with all other insertions. This can be attributed to the amount of artificial perilymph that was displaced in the tube of the cochlear model during insertion. Insertion at 0.1 mm/s displaced the fluid and raised the pressure in the cochlear model more slowly than any other insertion speed. Moreover, it was observed that the pressure at 0.1 mm/s increased more slowly with the insertion of an uncoated electrode compared with a coated one. Fittingly, the pressure plateau at this speed was also reached at a later insertion distance (uncoated vs. coated: 16 mm vs. 14 mm). Because the PLLA layer increased the diameter of the electrode array, the artificial perilymph was displaced from the model faster in comparison with the uncoated electrode array. However, no significant more rapid pressure increase was observed compared with uncoated electrode arrays (Table 1). Therefore, the coating was not considered to affect the surgical handling at an insertion speed of 0.1 mm/s. After all, a change in the pressure increase probably influenced the insertion process and could lead to a change in the insertion speed or the insertion force. The measured negative insertion pressures (−0.0036 and −0.0026 mmHg) were caused by the minimal fluctuation of the pressure measurement at the beginning. Since the pressure sensor is very sensitive, normal pressure fluctuations in the environment are registered.

The insertion pressure generated by the insertion of an uncoated electrode array was shown to be dependent on the insertion speed (Figure 3b,c). The maximum pressure increased as the insertion speed rose (Figure 3b). Similar results were shown by Todt et al. [45], who used a cochlear-scale model to show that the maximum insertion pressure grows as the insertion speed is increased. In contrast, the insertion pressure generated during insertion of a coated electrode array did not depend on the insertion speed (Figure 3e), but it was generally higher compared with that of the uncoated electrode arrays (Figure 4a).

The effect of faster displacement of the artificial perilymph explained the general increase in the maximum insertion pressure that occurred during the insertion of a coated electrode (Figure 4a). However, this effect seemed to have an influence only on the lower insertion speeds of coated electrodes. While there were detectable significant differences between the maximum pressures of the uncoated and coated arrays at 0.1, 0.5, and 1.0 mm/s, the pressure equalized at 1.5 mm/s and above. Consequently, from an insertion speed of 1.5 mm/s, the coating no longer seemed to have any influence on the maximum insertion pressure.

When considering the pressure created during the entire insertion, the uncoated and coated electrode arrays only differed significantly at an insertion speed of 0.5 mm/s. Therefore, the coating had hardly any influence when considering this parameter. This indicates that the coated electrode array behaved like an uncoated array during pressure development in the cochlea and even counteracted the negative effect caused by the increased diameter due to the PLLA coating from an insertion speed of 1.0 mm/s onward.

During CI implantation, the perilymph usually acts as a lubricant for the electrode array, and the endosteum lining of the scala tympani acts as a friction partner. To simulate these clinical conditions, a custom-made pulling device was used. A porcine endothoracic fascia was used to mimic the endosteum lining, as both are mainly composed of the same tissue type [48]. Moreover, artificial perilymph was used as a lubricant. The applied moving speeds were identical to the insertion speed used in cochlear implant studies [17,19,46,47].

The measurement of the insertion friction showed that the insertion speed had no significant influence on the friction coefficient of the coated electrode when moving alongside the mimic endosteum lining (Figure 5a). This was in contrast to the friction coefficient of the electrode array without coating. Here, an increased friction coefficient occurred with increasing movement speed (Figure 5b), similar to the description by Dohr et al. [18]. Generally, it was shown that the friction coefficient increased due to the coating compared with an uncoated electrode array (Figure 5b). Possibly, there is a similar relation between the insertion speed and friction coefficient that occurs at speeds lower than 0.1 mm/s only. Moreover, similar to the obtained maximum insertion pressure, the PLLA coating seemed to have an influence on the friction coefficient only at lower insertion speeds for the coated electrode arrays.

While there were detectable significant differences between the friction coefficient of the uncoated arrays at 0.1 and 0.5 mm/s, the pressures equalized at 1.0 mm/s and above in comparison with the uncoated arrays. Consequently, from an insertion speed of 1.0 mm/s, the coating no longer seemed to have any influence on the friction coefficient (Figure 5b).

Furthermore, the obtained increased friction coefficient at lower insertion speeds could also be a reason why the maximum insertion pressure increased at these insertion speeds. The wall friction was minimal, but there was certain contact with the model. This was also shown by the simultaneously recorded force data. Indeed, next to the insertion pressure, the insertion forces of the electrode arrays were also recorded. However, due to the linear orientation of the model and thus low wall contact of the electrode array, the maximum insertion forces did not exceed 0.015 N (data not shown). Since these insertion forces were very small compared with the maximum forces generated in a scale model within our own studies [18] (highest value 0.025 ± 0.007 N), a linear model is not suitable for representing the insertion forces. Thus, this parameter was not used for analysis.

To evaluate the influence of the mechanical stress during the pressure and friction experiments on the PLLA coating, SEM images of the selected electrodes were obtained after the experiments (Figure 6). The electrodes that produced the highest values for the maximum pressure (Figure 3e, 2.0 mm/s) and friction coefficient (Figure 5a, 1.5 mm/s) in the previous measurements were used for the SEM images. The coating of the electrode array used for the pressure measurement (Figure 6c–f) showed significant changes in its morphology compared with the control array (Figure 6a–c). Especially around the electrode contacts (Figure 6e), but also between the contacts (Figure 6d), noticeable wrinkling was observed. This can be attributed to the fact that the electrode array was inserted into the linear model as deep as possible during each insertion, thus compressing and crimping the coating. In addition, the coating was frequently subjected to high stress due to the 10-fold insertion of the electrode array.

Compared with the coating condition after the pressure measurement (Figure 6g–i), the coating after the friction experiment (Figure 6d–f) showed deep cracks only at the level of the electrode contacts (Figure 6h). The coating appeared to be more robust in the pulling motion (Figure 6g–i) than in the pushing motion (Figure 6d–f). However, within the friction measurement, the electrode was reused only half as often (5 times) as the electrode during the pressure measurement (10 times). In addition to the cracks at the level of the electrode contacts, rips and exfoliation could also be detected on the tip of the electrode array (Figure 6i). During the friction test, the electrode was screwed into a holder with the proximal end (electrode tip). Due to this, high mechanical forces acted on the end of the array and consequently on the coating. This fact explains the cracking and peeling of the coating at this part.

The silicon sheeting was interrupted for the current spreading around the electrode contacts. There, the optimal surface tension of the PLLA coating is interrupted as well. This may lead to cracks in the coating during the insertion process, where the electrode is bent along the radius of the deflection pulley or minor movements occur during the linear feed. The fine cracks in the coating around the contacts seemed to serve as a predetermined breaking point of the coating, because after both the pressure and friction experiments, increased damage to the coating was observed at this point. The cracking could become more relevant during the insertion process under real insertion conditions in the human cochlea.

## 5. Conclusions

This study is the first to investigate the insertion pressure and friction coefficient of PLLA-coated electrode arrays at different insertion speeds (0.1–2.0 mm/s) and compares the data with those of uncoated electrode arrays. This study shows that both the maximum insertion pressure generated in the linear model and the friction coefficient of a PLLA-coated electrode did not depend on the insertion speed. At higher insertion speeds, the maximum insertion pressure (1.5 mm/s and higher) and the friction coefficient (from 1.0 mm/s and higher) of the coated electrode array were similar to those of an uncoated array. When considering the pressure developed during the insertion, it was observed that there was no difference in the pressure development caused by a PLLA-coated electrode array compared to an uncoated electrode array, except at an insertion speed of 0.5 mm/s. Thus, it seems that a PLLA-coated electrode array acted like an uncoated electrode array at the mechanical parameters tested here (insertion pressure and insertion friction) when higher insertion speeds were applied. In general, the insertion of electrode arrays into the human cochlea in the course of CI implantation usually takes place at higher speeds (from 1.0 mm/s) [19]. In this case, PLLA coating does not cause any limitations at higher insertion speeds under the established parameters within this study (examining insertion pressure, friction, and use of the linear cochlear model). Thus, PLLA can be considered a suitable material to be used as a coating for electrodes and therefore as a potential drug delivery system. Future studies are considered in order to test the mechanical properties of drug-loaded PLLA coatings and improve the flexibility to allow insertion into a scale model or, in the long term, into human cochlea.

## Figures and Tables

**Figure 1 materials-15-03049-f001:**
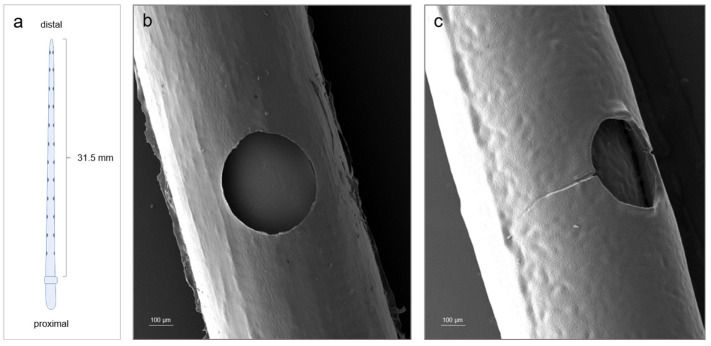
Investigated electrode arrays within this study. (**a**) Schematic illustration of a standard cochlear electrode array (MED-EL). Electrode array insertion part is 31.5 mm long. Ends of the electrode array are distinguished into proximal and distal ends. (**b**,**c**) SEM images of the fifth platinum electrode contact (hole, counted from distal to proximal end) and the surrounding area of the electrode array. (**b**) SEM image of an uncoated standard cochlear electrode array (MED-EL). (**c**) SEM image of a PLLA-coated standard cochlear electrode array (MED-EL).

**Figure 2 materials-15-03049-f002:**
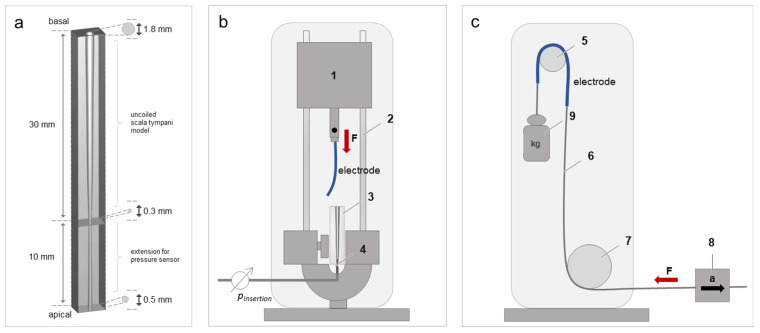
Schematic illustration of the test benches used to analyze the biomechanical properties of a standard cochlear electrode array. (**a**) Linear cochlea model separated into uncoiled scala tympani model (30 mm) and the extension intended for the pressure sensor (10 mm). The diameters range from 1.8 mm to 0.3 mm (uncoiled scala tympani) and 0.3 mm to 0.5 mm (extension) for the basal to apical openings, respectively. (**b**) Illustration of the insertion force test bench. Insertion pressure test bench comprises the load cell (1), linear feed (2), linear cochlear model (3), cochlea electrode array (electrode), and pressure sensor (4). (**c**) Illustration of the friction force test bench. Friction force test bench comprises the deflection cylinder covered with friction partner (5), wire rope (6), deflection pulley (7), load cell (8), counterweight (9), and cochlear electrode array (electrode). Red arrow with “F” indicates the recorded direction of the friction force, and black arrow with “a” indicates the pulling direction.

**Figure 3 materials-15-03049-f003:**
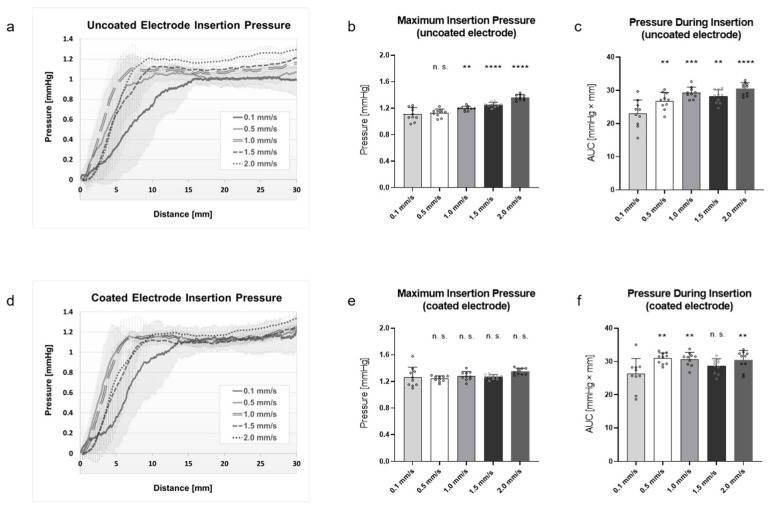
Cochlear electrode array insertion pressure measured with the linear cochlea model. Measurements were performed over a 30-mm insertion depth at 5 different insertion speeds (0.1, 0.5, 1.0, 1.5, and 2.0 mm/s). (**a**,**d**) Curve diagrams of insertion pressure of uncoated (**a**) and coated (**d**) electrode arrays. (**b**,**e**) Maximum insertion pressure of uncoated (**b**) and coated (**e**) electrode arrays. (**c**,**f**) Insertion pressure during insertion (area under the curve (AUC)) of uncoated (**c**) and coated (**f**) electrode arrays. n = 10 independent repetitions were performed for each insertion speed. Data are presented as mean with positive standard deviation (lines). Dots represent independent measured values. *p*-values were calculated by one-way ANOVA followed by Bonferroni’s multiple comparison post hoc test. ** *p* < 0.01. *** *p* < 0.001. **** *p* < 0.0001. n. s. = not significant. Data were compared to data at 0.1 mm/s.

**Figure 4 materials-15-03049-f004:**
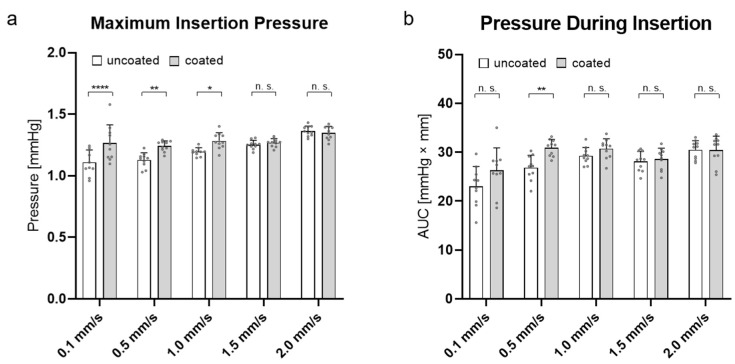
(**a**) Maximum insertion pressure of uncoated and coated electrode arrays at 5 different insertion speeds (0.1, 0.5, 1.0, 1.5, and 2.0 mm/s). (**b**) Insertion pressure during insertion (area under the curve (AUC)) of uncoated and coated electrode arrays at five different insertion speeds. n = 10 independent repetitions were performed for each insertion speed. Data are presented as mean with positive standard deviation (lines). Dots represent independent measured values. *p*-values were calculated by two-way ANOVA followed by Sidak’s multiple comparison post hoc test. * *p* < 0.05. ** *p* < 0.01. **** *p* < 0.0001. n. s. = not significant.

**Figure 5 materials-15-03049-f005:**
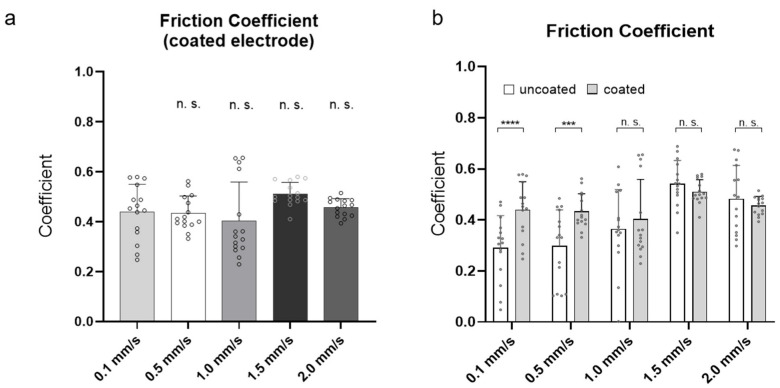
Friction coefficient of cochlear electrode arrays recorded during moving along a friction partner. Measurements were performed over a distance of 30 mm at 5 different moving speeds (0.1, 0.5, 1.0, 1.5, and 2.0 mm/s). (**a**) Friction coefficient of coated electrode arrays. (**b**) Comparison of friction coefficients of uncoated with coated electrode arrays. n = 15 independent repetitions were performed for each insertion speed. Data are presented as mean with positive standard deviation (lines). Dots represent independent measured values. P-values were calculated by one-way ANOVA followed by Bonferroni’s multiple comparison post hoc test (**a**) and two-way ANOVA followed by Sidak’s multiple comparison post hoc test (**b**). *** *p* < 0.001. **** *p* < 0.0001. n. s. = not significant (in (**a**), data were compared to mean friction coefficient at 0.1 mm/s).

**Figure 6 materials-15-03049-f006:**
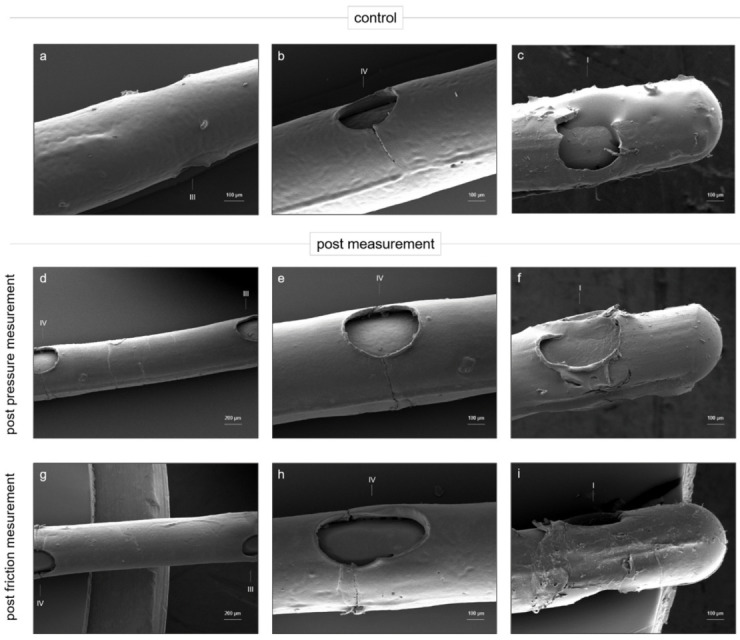
SEM images of PLLA-coated standard cochlear electrode arrays. (**a**–**c**) SEM images of PLLA-coated standard cochlear electrode array before measurement (control). (**d**–**f**) SEM images of PLLA-coated standard cochlear electrode array post pressure measurement. The electrode array was inserted 10 times into linear cochlea model at an insertion speed of 2.0 mm/s. (**g**–**i**) SEM images of PLLA-coated standard cochlear electrode array post friction measurement. The electrode array was moved 5 times along friction partner at a speed of 1.5 mm/s. (**a**,**d**,**g**) SEM image of PLLA-coated standard cochlear electrode array between two electrode contacts. (**b**,**e**,**h**) SEM image of PLLA-coated standard cochlear electrode array around the fourth electrode contact. (**c**,**f**,**i**) SEM image of the tip of the PLLA-coated standard cochlear electrode array. The electrode contacts are numbered with roman numerals (I, III, and IV).

**Table 1 materials-15-03049-t001:** Slope (m) of insertion pressure of uncoated and coated electrode arrays at different insertion speeds (0.1, 0.5, 1.0, 1.5, and 2.0 mm/s). *p*-values were calculated by two-way ANOVA followed by Sidak’s multiple comparison post hoc test. ** *p* < 0.01. *** *p* < 0.001. Data were compared to data at 0.1 mm/s of the respective column.

Speed (mm/s)	m (Uncoated) (mmhg/mm)	m (Coated) (mmhg/mm)
0.1	0.11 ± 0.04	0.13 ± 0.04
0.5	0.18 ± 0.05	0.23 ± 0.05 ***
1.0	0.22 ± 0.07 ***	0.19 ± 0.06 **
1.5	0.18 ± 0.05	0.19 ± 0.07
2.0	0.20 ± 0.06 **	0.19 ± 0.06

## Data Availability

The raw data required to reproduce these findings cannot be shared at this time as the data also forms part of an ongoing study.

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
