# Peer review of "A PLLA Coating Does Not Affect the Insertion Pressure or Frictional Behavior of a CI Electrode Array at Higher Insertion Speeds"

_materials, 2022, doi:10.3390/ma15093049_

Round 1
Reviewer 1 Report
See attached pdf.

Author Response
Dear Reviewer,
thank you very much for taking the time to critically review our work.
We hope that we have satisfactorily addressed all of your useful comments and suggestions. Please see the attachment.
Sincerely,
Dana Dohr

Reviewer 2 Report
The manuscript has a novel theme and is worth publishing, with some detailed modifications suggested.
1.The conclusion part (P15) “…In this case, a PLLA coating does not create any limitations in the parameters set at higher insertion speeds…”: It is questionable that such conclusion is given only based on the pressure and frictional force of an in-vitro linear model of cochlea. Is it possible to provide the data of other models or in-vivo models to support the conclusion?
2.The author only tested a coating thickness (10 µm). Can the author explain why this thickness is selected? Do other PLLA coating’s electrode arrays also follow the same law in terms of insertion pressure and frictional force?
3.According to the SEM graphs on P11, the PLLA coating will deform and produce bits after the pressure and friction tests. Are there any possible safety issues arising from those bits in the applications?
Author Response

(The authors gave the same response as above.)
